# Numerical Evaluation of Potential Catalyst Savings for Ventilation Air Methane Catalytic Combustion in Helical Coil Reactors with Selective Wall Coating

**Jundika C. Kurnia** [1], **Benitta A. Chaedir** [2], **Desmond C. Lim** [1], **Lianjun Chen** [3], **Lishuai Jiang** [3] **and Agus P. Sasmito** [2],*

[1] Department of Mechanical Engineering, Universiti Teknologi PETRONAS, 32610 Bandar Seri Iskandar, Perak Darul Ridzuan, Malaysia; jundika.kurnia@utp.edu.my (J.C.K.); desmond_17005787@utp.edu.my (D.C.L.)

[2] Department of Mining and Materials Engineering, McGill University, Frank Dawson Adams Bldg., 3450 University Street, Montreal, QC H3A2A7, Canada; benitta.chaedir@mail.mcgill.ca

[3] State Key Laboratory of Mining Disaster Prevention and Control, Shandong University of Science and Technology, Qingdao 266590, China; skyskjxz@163.com (L.C.); jlsh1989@126.com (L.J.)

* Correspondence: agus.sasmito@mcgill.ca; Tel.: +1-514-398-3788

**Abstract:** During active mining operation of a gassy underground mine, large amounts of methane will be released from the mine ventilation shaft. To eliminate the harmful effects of this ventilation air methane and minimize the wastage of this potential energy resource, considerable effort has been devoted to converting this alternative fuel using catalytic combustion. This study numerically investigated the reaction performance of ventilation air methane (VAM) in helical coil tubes of various configurations utilizing a computational fluid dynamics (CFDs) approach. Several key factors affecting the catalytic combustion performance such as curvature, inlet Reynolds number, and cross-section aspect ratio were evaluated. Recalling the high cost of the catalyst used in this reaction—platinum—optimization of catalyst usage by implementing selective catalyst coating was conducted and investigated. For evaluation purposes, the reaction performance of the helical coil tube was compared to its straight counterpart. The results gave a firm confirmation of the superior performance of the helical coil tube compared to the straight one. In addition, it was found that the selective inner wall coating in the circular cross-section at a higher Reynolds number gave rise to the highest figure of merit (FoM), defined as the net energy produced per mg of catalyst platinum.

**Keywords:** catalytic combustion; ventilation air methane; helical reactor; selective coating

## 1. Introduction

For decades, methane has been considered as one of the main hazards in underground mines that needs careful attention and treatment, especially in mines where a high methane flowrate is emitted from the coal deposits (gassy mine). It is continuously produced in the active mining area during the mineral deposit excavation. This methane emission, if not treated properly, will accumulate in the mining face, posing risks for the mining personnel and mining operation, sometimes leading to fatal incidents [1,2]. Generally, large ventilation systems will be installed to supply excess fresh air and disperse this methane emission in order to maintain the safe operational condition for the miner [3]. The contaminated air will then be directed to the main ventilation shaft and pumped to the surface. Often, this diluted methane, commonly referred to as ventilation air methane (VAM), will be directly discharged into the atmosphere as it is deemed to have no economic value due to its low methane concentration (mostly below 1%) [4]. Nevertheless, there is a growing concern that direct release of

VAM into the atmosphere may contribute to the alarming accumulation of greenhouse gases in the atmosphere and is a wastage of an alternative fuel source. Therefore, several methods and technologies have been proposed and developed to recover and utilize VAM from the gassy mine [5].

The low concentration of methane makes combustion mitigation more challenging. Traditional combustion technologies as used in carbon dioxide mitigations, such as low NOx combustion and oxycombustion, may not be able to provide sufficient thermal energy to ignite and sustain the reaction. Moderate or intense low-oxygen dilution (MILD) combustion is one of the alternative technologies that may be considered for ultra-lean methane combustion, as it has the potential to oxidize low concentration fuel mixtures. The technology offers advantages in terms of improved energy efficiency and reduced pollutant emissions [6–9]. In MILD combustion, the temperature increase in the combustion process is lower than the mixture auto-ignition temperature, due to the high temperature of reactant mixture and the high-level dilution [6,7]. Moderate or intense low-oxygen dilution combustion involves mixing reactants and pre-heating the mixture, which essentially affects the thermodynamic properties and sustainability of the combustion process. Studies on MILD combustion performance and the effects of parameters, such as fuel and thermal load, on the combustion stability have been conducted. Recently, Sabia et al. [7] investigated the MILD combustion performance in a cyclonic burner as affected by mixture equivalence ratio and thermal power. It was found that the system best performs at an equivalence ratio slightly lower than one and a well identified operating temperature range for minimal pollutant emissions. The investigation of the performance and stability characteristics of MILD combustion was later extended to the undiluted and non-preheated air by Sorrentino et al. [8]. The results showed that the stable combustion process can be achieved under a wide range of operating conditions, even when it is fed with undiluted air or non-heated air. The process was able to self-stabilize in the absence of pre-heating and demonstrated good performance with complete fuel conversion and low pollutant emissions. Despite the great potential of the MILD (flameless) combustion, the technology has limited industrial applications and carries additional cost due to the need for preheating. Furthermore, the underlying physicochemical mechanisms of the system has yet to be fully investigated.

Catalytic reaction is widely adopted in many fields and applications, ranging from hydrogenation [10], semi-hydrogenation [11], chemical synthesis [12], to pollutant (such as $CO_2$ and methane) mitigation. A detailed review on the methods and applications of heterogenous catalysis is provided by Tanimu et al. [13]. In the case of ventilation air methane (VAM), its reutilization technologies can be divided into two main categories, i.e., ancillary uses and principal uses [14,15]. The former refers to utilization of VAM as combustion air for boilers, turbines, and internal combustion engines. The latter is related to the usage of VAM as the main fuel to produce energy through catalytic oxidation. Various technologies which use VAM as a principal fuel have been studied, including thermal and catalytic flow reserve reactors, catalytic-monolith reactors, lean burn gas turbines, and concentrators. Both thermal flow-reversal reactor (TFRR) and catalytic flow-reversal reactor (CFRR) technologies have the same operating principles. They employ flow-reversal principles to transfer heat from methane combustion to the incoming air so as to raise ventilation air temperature and promote methane ignition [16]. Catalytic flow-reversal reactors differ from the TFRR only with respect to the use of a catalyst in the CFRR technology. The CFRR decreases the autoignition temperature of methane in ventilation air, resulting in operational advantages [17]. However, the amount of heat generated increases with the concentration of methane in ventilation air, and too much heat will degrade the catalyst. The CFRR also requires a minimum methane concentration to maintain the reaction, which makes it more expensive due to the potential need for replacement of the catalysts [6]. The main limitation of the flow reversal technologies is in the difficulty to extract useful energy for power generation—heat recovered must be transferred into a working fluid [15]. Catalytic-monolith reactors (CMR) utilize a monolith honeycomb reactor containing hundreds of parallel channels. This honeycomb-type monolithic structure is superior for its outstanding characteristics of very low pressure drop at high mass throughputs and high mechanical strength [18].

The TFRR, CFRR, and CMR technologies are promising VAM mitigation methods and have received widespread attention. Detailed two-dimensional reverse-flow methane combustion model was solved by Aube and Sapoundjiev [19] for the prediction of the transient behaviour of laboratory-scale flow reversal reactors using a finite-volume method. The model was validated with the experimental results of oxidation of lean methane emissions. It was able to accurately predict the dynamic behaviour of the CFRR even for small reactor diameters and low air flow rates by taking into account the radial effects related to the thermal insulation. Gosiewski [20] performed one-dimensional simulations for the catalytic combustion of methane in a reverse-flow reactor for a manganese and palladium catalyst. It was found that high heat recovery can be obtained only at the expense of high catalyst temperature. However, high temperature may promote homogeneous combustion which can deactivate the catalyst. The effects of initial temperature, cycle time, feed gas concentration, and space velocity on catalytic combustion of VAM in a vertical reactor were studied by Wang et al. [4]. Experimental results showed that the reactor could run under a wide range of operating conditions with self-sustaining operation and high methane conversion, in the absence of CO and $NO_x$ generation. A three-dimensional model was developed and simulated by Lan and Li [21] in an effort to understand the thermodynamic characteristics of the thermal oxidation of methane in a TFRR. They also analyzed the effects of channel length, feed methane concentration, inlet velocity, and cycle time on the reactor behavior. It was observed that the lowest feed methane concentration for self-maintained running decreases with increasing channel length and rises significantly with inlet velocity, while cycle time has no effect. Mei et al. [22] numerically studied the performances of catalytic combustion of $CH_4$/air in a single channel and whole monolith reactor. The reactor was assumed to be cylindrical with many axially parallel channels, of which the arrangement was a correctitude triangle in the cross-section of the reactor, and the catalyst was dispersed in the washcoat coated onto the surface of the channels. The results showed that the simulations based on the whole reactor gave more relevant results and should be adopted to get a better insight into combustion performance. Analysis of monolith reactors with different structural parameters and simulation conditions showed that reaction rates increase with increasing specific surface area, while keeping voidage constant. Experimental and numerical studies of catalytic methane combustion in a honeycomb catalytic monolith burner were carried out by Dupont et al. [23]. The monoliths studied contained various concentrations of Pt or Pd on an alumina-based washcoat over a cordierite honeycomb support. The palladium catalysts were found to allow lower $CH_4$ concentrations with stable operation more than the platinum catalysts. For both long and short monoliths, the combustion occurred until completion, with no detectable CO, $NO_x$, or unburned fuel, i.e., near-zero pollutant emissions.

On these oxidation combustion studies for VAM, one key parameter affecting the reaction performance was the catalyst used in the reaction. There are two types of catalysts commonly used for VAM catalytic combustion, i.e., noble metal catalysts and metal oxide catalysts. The former is preferred as it offers higher catalytic activity, lower ignition temperature, and better anti-poisoning ability [4]. It is worth noting, however, that this noble metal catalyst is relatively expensive. Hence, it should be used wisely to achieve the optimum reaction rate. One way to realize this is by changing the flow geometry which can enhance the reaction rate using swirl flow [24], cyclonic flow [25], or by using a curved design reactor. Curved design (including coiled tubes) has been widely adopted to achieve higher heat and mass transfer as well as better performance mixing and reaction [26]. This curved design has also been implemented as a micro-combustor for thermo-photovoltaic application [27].

Selective coating is another method to reduce catalyst usage, and hence, catalyst cost. This strategy has been widely studied to ensure that the implementation of partial coating does not impair the reaction performance. For example, Di Benedetto et al. [28] numerically investigated catalytic combustion of methane in a monolith reactor through simulations using a computational fluid dynamics (CFD) approach. The authors studied the reactor behavior by first removing catalysts from two inner-channels, leaving the catalysts over the walls of all the other channels, then further removing catalysts from the walls of the other channels, except from the four external channels. The results showed complete

methane conversion achieved in both cases, verifying the performance of a partially-coated monolith reactor. Following this work, the reaction behavior was then studied in more detailed by Landi et al. [29] and Di Sarli et al. [30]. At the beginning, homogeneous reaction in the coated external channels was stimulated by heating the external channels from the environment. As the temperature of the inner channels increased due to heat conduction, the reaction in these uncoated inner channels was activated, thus allowing homogeneous reaction to occur throughout the entire monolith. On the basis of these numerical results [28–30], a novel partially coated monolith has been proposed and successfully tested by experiment [31]. Selective coating applied in the catalytic combustion in a micro-combustor was examined by Di Benedetto et al. [32] using a two-dimensional CFD model. In the study, the first part of the combustor was coated with catalysts, while a catalyst was absent in the second part. It was shown that the reaction was sustained even in the non-coated part, owing to the heat diffusing from the upstream catalytic section.

Our previous study [33] indicated that coiled tubes offer better reaction at shorter length compared to its straight counterpart. In our following study [34], we found the potential savings of a catalyst of heterogeneous reaction of methane oxidation in spiral coiled reactors by applying selective coating. The results indicated the potential application of this method for process intensification and catalyst cost reduction. In the present study, we extended our study to evaluate the potential catalyst savings of VAM catalytic combustion in a curved reactor with selective wall coating using computational fluid dynamics (CFDs) approach. The evaluated selective coatings are inner wall coating, outer wall coating, top wall coating, bottom wall coating, and all wall coating. In addition, the effect of reactor length and curvature radius will be evaluated with the main objective being to obtain the most favourable geometry and operating conditions to achieve optimum reaction performance while maintaining minimum cost in term of catalyst usage and required pumping power. For evaluation purposes, the VAM reaction performance in a straight reactor will be used as a benchmark.

## 2. Mathematical Model

In this study, helical coil reactors with various cross-section geometries were considered, similar to our previous study [35]. These geometries are illustrated in Figure 1. Similar to our previous studies [33,34], several assumptions were taken when developing this model, i.e., premix inlet condition, reaction occurs at the reactor wall, Newtonian fluid steady, and laminar flow. Detailed geometric and operatic parameters are summarized in Table 1.

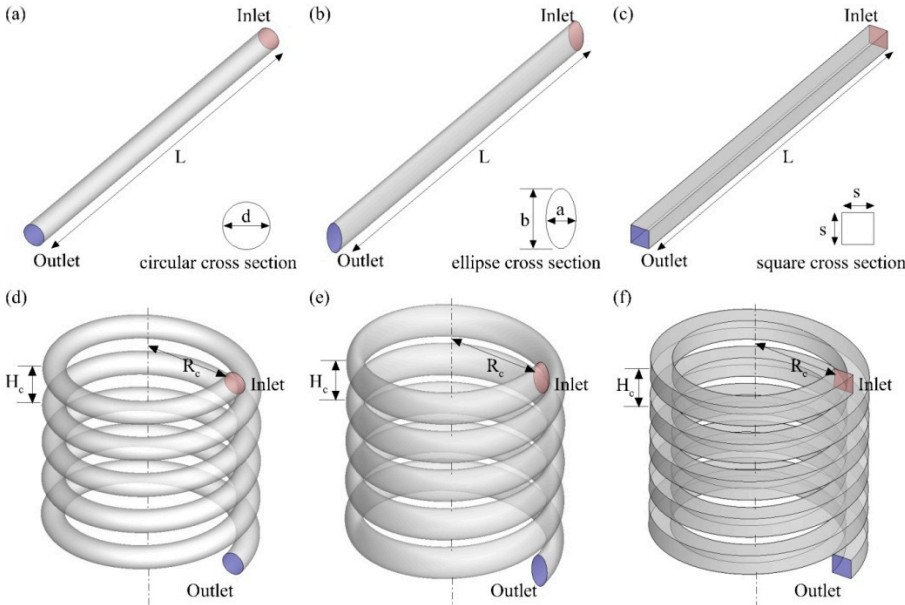

**Figure 1.** Schematics of the investigated cooling channel.

**Table 1.** Geometric parameters, operating parameters, and material properties.

| Parameter | Symbol | Value | Unit |
|---|---|---|---|
| Diameter of circular cross-section | $d$ | $1.13 \times 10^{-3}$ | m |
| Side of square cross-section | $s$ | $1.00 \times 10^{-3}$ | m |
| Minor axis of ellipse cross-section | $a$ | $7.98 \times 10^{-4}$ | m |
| Major axis of ellipse cross-section | $b$ | $1.60 \times 10^{-3}$ | m |
| Pitch/distance between helical turn | $h$ | $2.00 \times 10^{-3}$ | m |
| Helical coil radius | $R_c$ | $4.00 \times 10^{-3}$ | m |
| Total length of reactor | $L$ | $12.58 \times 10^{-2}$ | m |
| Platinum coverage on the surface | Pt (s) | $2.71 \times 10^{-8}$ | kmol m$^{-2}$ |
| Inlet velocity | $u^{in}_{mixture}$ | 1.38, 6.88,13.77, 20.65 (square) 1.50, 7.51, 15.03, 22.54 (ellipse) 1.56, 7.78, 15.56, 23.33 (square) | m s$^{-1}$ |
| Inlet oxygen mass fraction | $\omega^{in}_{O_2}$ | 0.23 | - |
| Inlet methane mass fraction | $\omega^{in}_{CH_4}$ | 0.01 | - |
| Inlet hydrogen mass fraction | $\omega^{in}_{H_2}$ | $4.50 \times 10^{-2}$ | - |
| Inlet temperature | $T^{in}_{mixture}$ | 300 | K |
| Wall temperature | $T_{wall}$ | 1290 | K |

## 2.1. Governing Equations

According to the taken assumption, the conservation equation of mass, momentum, energy, and species inside the reactor can be expresses as [33,34]

$$\nabla \cdot \rho \mathbf{u} = 0, \tag{1}$$

$$\nabla \cdot (\rho \mathbf{u} \mathbf{u}) = -\nabla p \mathbf{I} + \nabla \cdot \left[ \mu \left( \nabla \mathbf{u} + (\nabla \mathbf{u})^T \right) - \frac{2}{3} \mu (\nabla \cdot \mathbf{u}) \mathbf{I} \right], \tag{2}$$

$$\nabla \cdot (\rho \mathbf{u} \omega_i) = \nabla \cdot (\rho D_i \nabla \omega_i) + R_i, \tag{3}$$

$$\nabla \cdot \left( \rho c_p \mathbf{u} T \right) = \nabla \cdot \left( k_{eff} \nabla T \right) + S_{temp}, \tag{4}$$

where $\rho$ is the fluid density, $\mathbf{u}$ the fluid velocity, $p$ the pressure, $\mu$ the fluid dynamic viscosity, $\omega_i$ and $D_i$ are mass fraction and diffusion coefficient of species $i$, $R_i$ the mass consumed or produced by the reactions at the catalyst coated wall, $c_p$ the specific heat of the gas mixture, $k_{eff}$ is the effective thermal conductivity, $T$ the temperature, and $S_{temp}$ the heat released/absorbed due to reaction.

## 2.2. Methane Catalytic Oxidation Reaction

Here, the reaction taken into account is the heterogeneous reaction of methane oxidation occurring at the platinum catalyst-coated wall of the reactor. The model distinguishes the chemical species deposited on the wall surface with those in the bulk flowing gas. Similarly, the reaction occurring on the wall surface for the species deposited on the wall is treated differently from the reaction in the bulk gas. The model considers one bulk/solid species (Pt(b)), seven gas species (CH$_4$, O$_2$, H$_2$, H$_2$O, CO, CO$_2$, and N$_2$), and 11 surface species (H(s), Pt(s), O(s), OH(s), H$_2$O(s), CH$_3$(s), CH$_2$(s), CH(s), C(s), CO(s), CO$_2$(s)) that illustrate the coverage of the surface with adsorbed species. The detailed multi-step reaction mechanism and its reactions rate constants are summarized in Table 2.

**Table 2.** Surface reaction mechanism.

| No. | Reaction | $A_r$ | $\beta_r$ | $E_r$ (J/kmol) |
|---|---|---|---|---|
| 1 | $H_2 + 2Pt(s) \rightarrow 2H(s)$ | $4.36 \times 10^7$ | 0.5 | 0 |
| 2 | $2H(s) \rightarrow H_2 + 2Pt(s)$ | $3.70 \times 10^{20}$ | 0 | $6.74 \times 10^7$ |
| 3 | $O_2 + 2Pt(s) \rightarrow 2O(s)$ | $1.80 \times 10^{17}$ | −0.5 | 0 |
| 4 | $O_2 + 2Pt(s) \rightarrow 2O(s)$ | $2.01 \times 10^{14}$ | 0.5 | 0 |
| 5 | $2O(s) \rightarrow O_2 + 2Pt(s)$ | $3.70 \times 10^{20}$ | 0 | $2.13 \times 10^8$ |
| 6 | $H_2O + Pt(s) \rightarrow H_2O(s)$ | $2.37 \times 10^8$ | 0.5 | 0 |
| 7 | $H_2O(s) \rightarrow H_2O + Pt(s)$ | $1.00 \times 10^{13}$ | 0 | $4.03 \times 10^7$ |
| 8 | $OH + Pt(s) \rightarrow OH(s)$ | $3.25 \times 10^8$ | 0.5 | 0 |
| 9 | $OH(s) \rightarrow OH + Pt(s)$ | $1.00 \times 10^{13}$ | 0 | $1.93 \times 10^8$ |
| 10 | $H(s) + O(s) \rightarrow OH(s) + Pt(s)$ | $3.70 \times 10^{20}$ | 0 | $1.15 \times 10^7$ |
| 11 | $H(s) + OH(s) \rightarrow H_2O(s) + Pt(s)$ | $3.70 \times 10^{20}$ | 0 | $1.74 \times 10^7$ |
| 12 | $OH(s) + OH(s) \rightarrow H_2O(s) + O(s)$ | $3.70 \times 10^{20}$ | 0 | $4.82 \times 10^7$ |
| 13 | $CO + Pt(s) \rightarrow CO(s)$ | $7.85 \times 10^{15}$ | 0.5 | 0 |
| 14 | $CO(s) \rightarrow CO + Pt(s)$ | $1.00 \times 10^{13}$ | 0 | $1.25 \times 10^8$ |
| 15 | $CO_2(s) \rightarrow CO_2 + Pt(s)$ | $1.00 \times 10^{13}$ | 0 | $2.05 \times 10^7$ |
| 16 | $CO(s) + O(s) \rightarrow CO_2(s) + Pt(s)$ | $3.70 \times 10^{20}$ | 0 | $1.05 \times 10^8$ |
| 17 | $CH_4 + 2Pt(s) \rightarrow CH_3(s) + H(s)$ | $2.30 \times 10^{16}$ | 0.5 | 0 |
| 18 | $CH_3(s) + Pt(s) \rightarrow CH_2(s) + H(s)$ | $3.70 \times 10^{20}$ | 0 | $2 \times 10^7$ |
| 19 | $CH_2(s) + Pt(s) \rightarrow CH(s) + H(s)$ | $3.70 \times 10^{20}$ | 0 | $2 \times 10^7$ |
| 20 | $CH(s) + Pt(s) \rightarrow C(s) + H(s)$ | $3.70 \times 10^{20}$ | 0 | $2 \times 10^7$ |
| 21 | $C(s) + O(s) \rightarrow CO(s) + Pt(s)$ | $3.70 \times 10^{20}$ | 0 | $6.28 \times 10^7$ |
| 22 | $CO(s) + Pt(s) \rightarrow C(s) + O(s)$ | $1.00 \times 10^{17}$ | 0 | $1.84 \times 10^8$ |
| 23 | $OH(s) + Pt(s) \rightarrow H(s) + O(s)$ | $1.56 \times 10^{18}$ | 0 | $1.15 \times 10^7$ |
| 24 | $H_2O(s) + Pt(s) \rightarrow H(s) + OH(s)$ | $1.88 \times 10^{18}$ | 0 | $1.74 \times 10^7$ |
| 25 | $H_2O(s) + O(s) \rightarrow OH(s) + OH(s)$ | $4.45 \times 10^{20}$ | 0 | $4.82 \times 10^7$ |

Surface reaction can produce and consume both gas-phase and surface species. This reaction can be expressed as

$$\sum_{i=1}^{N_g} g'_{i,r} G_i + \sum_{i=1}^{N_b} b'_{i,r} B_i + \sum_{i=1}^{N_s} s'_{i,r} S_i \overset{K_r}{\leftrightarrow} \sum_{i=1}^{N_g} g''_{i,r} G_i + \sum_{i=1}^{N_b} b''_{i,r} B_i + \sum_{i=1}^{N_s} s''_{i,r} S_i. \tag{5}$$

In Equation (5), $G_i$, $B_i$, and $S_i$ represent the gas-phase, solid species, and the surface-adsorbed species, respectively. Meanwhile, $g'$, $b'$, and $s'$ are the stoichiometric coefficients for each reactant species; $g''$, $b''$, and $s''$ are the stoichiometric coefficients for each product species; and $K_r$ is the overall reaction rate constant. Based on the condition that only the species involved as reactants or products will have a non-zero stoichiometric coefficient, the rate of reaction is defined as

$$\mathfrak{R} = k_{f,r} \prod_{i=1}^{N_g} [G_i]_{wall}^{g'_{i,r}} [S_i]_{wall}^{s'_{i,r}}, \tag{6}$$

where $[G_i]_{wall}^{g'_{i,r}}$ denotes molar concentration on the wall which can be calculated as

$$[G_i]_{wall} = \frac{\rho_{wall} \omega_{i,wall}}{M_{w,i}}, \tag{7}$$

while $[S_i]_{wall}$ is the site species concentration at the wall, which is given by

$$[S_i]_{wall} = \rho_{site} z_i, \tag{8}$$

where $\rho_{site}$ is the site density of the catalyst and $z_i$ is the site coverage of species $i$.

The net molar rate of production or consumption of each species $i$ can therefore be estimated as

$$\hat{R}_{i,gas} = \sum_{r=1}^{N_{rxn}} (g''_{i,r} - g'_{i,r})\mathfrak{R}_r \quad (i = 1, 2, 3, \ldots, N_g),$$

$$\hat{R}_{i,bulk} = \sum_{r=1}^{N_{rxn}} (b''_{i,r} - b'_{i,r})\mathfrak{R}_r \quad (i = 1, 2, 3, \ldots, N_b), \tag{9}$$

$$\hat{R}_{i,site} = \sum_{r=1}^{N_{rxn}} (s''_{i,r} - s'_{i,r})\mathfrak{R}_r \quad (i = 1, 2, 3, \ldots, N_s).$$

Meanwhile, the reaction rate constant is given by

$$k_{f,r} = A_r T^{\beta_r} \exp\left(-\frac{E_r}{RT}\right). \tag{10}$$

Reactions at surfaces alter gas-phase, surface-adsorbed (site) and bulk (solid) species. At the catalyst-coated wall surfaces, the mass flux of each gas species due to diffusion and convection to or from the surface is counter-balanced with its rate of consumption/production on the surface

$$\rho_{wall} D_i \frac{\partial \omega_{i,wall}}{\partial n} - \dot{m}_{dep} \omega_{i,wall} = M_{w,i} \hat{R}_{i,gas} \quad (i = 1, 2, 3, \ldots, N_g), \tag{11}$$

$$\frac{\partial [S_i]_{wall}}{\partial t} = \hat{R}_{i,site} \quad (i = 1, 2, 3, \ldots, N_s). \tag{12}$$

where $\dot{m}_{dep}$ is the net rate of mass deposition or etching as a result of surface reaction, i.e.,

$$\dot{m}_{dep} = \sum_{i=1}^{N_b} M_{w,i} \hat{R}_{i,bulk}, \tag{13}$$

The diffusion term in Equations (11) and (12) is computed as the ratio of the difference in the mass fraction of species at the cell center and the wall-face center to the normal distance between these center points. A point-by-point coupled Newton solver is utilized to solve Equations (10) and (11) for the dependent variables $\omega_{i,wall}$ and $z_i$. If the Newton solver cannot solve these equations, time marching in an ordinary differential equations (ODE) solver is used until convergence is reached. In the condition where the ODE solver cannot solve these equations, reaction–diffusion balance will be disabled, $\omega_{i,wall}$ is assumed equal to the cell-center value, $\omega_{i,cell}$, and only the site coverage $z_i$ are advanced in the ODE solver to convergence.

### 2.3. Constitutive Relations

In this study, the gas mixture is treated as ideal gas. Thus, the density can be calculated by using ideal gas law, i.e.,

$$\rho = \frac{PM}{R_u T}, \tag{14}$$

where $R_u$ is the universal gas constant. The mixture molar mass, $M$ can be expressed as

$$M = \left(\frac{\omega_{CH_4}}{M_{CH_4}} + \frac{\omega_{H_2}}{M_{H_2}} + \frac{\omega_{O_2}}{M_{O_2}} + \frac{\omega_{H_2O}}{M_{H_2O}} + \frac{\omega_{CO_2}}{M_{CO_2}} + \frac{\omega_{CO}}{M_{CO}} + \frac{\omega_{N_2}}{M_{N_2}}\right)^{-1}. \tag{15}$$

Here, $M_i$ is the molar mass of species $i$.

Meanwhile, the gas mixture viscosity ($\mu$) can be calculated by using averaging method, i.e.,

$$\mu = \sum_\alpha \frac{x_\alpha \mu_\alpha}{\sum_\beta x_\beta \Phi_{\alpha,\beta}}, \quad (\alpha, \beta = CH_4, H_2, O_2, H_2O, CO, CO_2, N_2), \tag{16}$$

where $x_{\alpha,\beta}$ are the mole fractions of species $\alpha$ and $\beta$. The coefficient $\Phi_{\alpha,\beta}$ is given by

$$\Phi_{\alpha,\beta} = \frac{1}{\sqrt{8}}\left(1 + \frac{M_\alpha}{M_\beta}\right)^{-1/2}\left[1 + \left(\frac{\mu_\alpha^{(g)}}{\mu_\beta^{(g)}}\right)^{1/2}\left(\frac{M_\beta}{M_\alpha}\right)^{1/4}\right]^2,$$

(17)

where $\mu_{\alpha,\beta}$ are the viscosity of individual species $\alpha$ and $\beta$.

Similarly, the mixture thermal conductivity, $k_{eff}$ and specific heat capacity, $c_p$ can be calculated by

$$k_{eff} = \sum k_i \omega_i,$$

(18)

$$c_p = \sum_i \omega_i c_{p,i}.$$

(19)

The performance of the reactor is evaluated and discussed in terms of local and overall reactant conversion, $\zeta$ and figure of merit (FoM) which are defined as

$$\zeta_{i,x} = \frac{\omega_{i,mean}^x - \omega_{i,mean}^{out}}{\omega_{i,mean}^{in}},$$

(20)

$$\zeta_i = \frac{\omega_{i,mean}^{in} - \omega_{i,mean}^{out}}{\omega_{i,mean}^{in}},$$

(21)

$$FoM = \frac{\left(\dot{E}_{combustion} - P_{load}\right)}{m_{catalyst}}$$

(22)

where $\omega_{i,mean}$ is the mixed mean mass fraction which is defined as

$$\omega_{i,mean} = \frac{1}{VA_c}\int_{A_c} \omega_i u\, dA_c,$$

(23)

and $V$ is the mean velocity which is given by

$$V = \frac{1}{A_c}\int_{A_c} u\, dA_c.$$

(24)

The power generated from methane catalytic combustion can be calculated as

$$\dot{E} = \eta_{pg}\left(\dot{m}_{CH_4,mean}^{in} - \dot{m}_{CH_4,mean}^{out}\right)\Delta H_{c,CH_4}$$

(25)

where $\eta_{pg}$ is the power generation efficiency, $\Delta H_{c,CH_4}$ is combustion enthalpy for methane, $\dot{m}_{CH_4,mean}^{in}$ and $\dot{m}_{CH_4,mean}^{out}$ are the methane mass flow rate at the inlet and outlet, respectively. Currently, VAM is utilized as fuel for steam power plant utilizing catalytic oxidation [36]. Typical steam power plant has an efficiency within 38% to 45% [37]. In this study, the efficiency of the power generation was taken at a conservative value 33%. Meanwhile, the enthalpy combustion of methane was taken as 890.7 kJ/mol [38,39]. The load power corresponds to the pumping power required to drive the VAM through the reactor, i.e.,

$$P_{pump} = \left(\frac{1}{\eta_{pump}}\right)\dot{V}\Delta P,$$

(26)

where $\eta_{pump}$ is the pump efficiency (taken as 80%), $\dot{V}$ is the volume flow rate, and $\Delta P$ is the pressure drop. The mass of catalyst can be calculated from the catalyst loading, $\rho_{site}$ and the coating area, as listed in Table 3.

**Table 3.** Catalyst area (in cm$^2$) for various coating strategies.

| Geometry | Reactor Length | | | |
|---|---|---|---|---|
| | **31 mm** | **63 mm** | **94 mm** | **126 mm** |
| Straight circle | 111.69 | 223.08 | 334.62 | 446.16 |
| Straight ellipse | 121.75 | 243.37 | 365.06 | 486.74 |
| Straight square | 125.86 | 251.36 | 377.04 | 502.72 |
| Helical circle inner | 50.84 | 101.68 | 152.53 | 203.37 |
| Helical circle outer | 60.85 | 121.70 | 182.55 | 243.40 |
| Helical circle all | 111.69 | 223.38 | 335.08 | 446.77 |
| Helical ellipse inner | 56.59 | 113.19 | 169.78 | 226.37 |
| Helical ellipse outer | 65.16 | 130.32 | 195.48 | 260.64 |
| Helical ellipse all | 121.75 | 243.50 | 365.25 | 487.01 |
| Helical square inner | 57.05 | 114.09 | 171.14 | 228.18 |
| Helical square outer | 68.82 | 137.63 | 206.45 | 275.27 |
| Helical square all | 125.86 | 251.73 | 377.59 | 503.45 |

### 2.4. Boundary Condition

The required boundary conditions to complete the developed model are as follows:

- Inlet: premix reactant is supplied to the reactor. The velocity, temperature, and species mass fraction are set,

$$u = u^{in}_{mixture}, T = T^{in}_{mixture}, \omega_{O_2} = \omega^{in}_{O_2}, \omega_{CH_4} = \omega^{in}_{CH_4}, \omega_{H_{24}} = \omega^{in}_{H_2},$$
$$\omega_{N_2} = 1 - \left(\omega^{in}_{O_2} + \omega^{in}_{CH_4} + \omega^{in}_{H_2}\right). \tag{27}$$

Inlet velocity for the mixture is summarized in Table 1.

- Reaction walls: The reaction takes place on the wall and is resolved by Equation (9). To activate the wall surface reaction, the reaction mechanism is incorporated to the species boundary condition. No-slip condition are applied. The wall temperature is not known a priori and needs to be iterated for from the heat source due to the reaction. Note that the initial temperature is very important in the steady state model to trigger the reaction.

$$\mathbf{u} = 0, T_{init} = T_{wall} \tag{28}$$

- Non-reaction walls: Similar to boundary conditions for the reaction wall but the reaction on these walls is turned off.
- Outlet: The pressure and stream-wise gradient of the temperature and species mass fraction is set to zero.

$$p = p_{out}, \ \mathbf{n} \cdot \nabla T = \mathbf{n} \cdot \nabla \omega_i = 0 \tag{29}$$

The inlet velocity adopted in this study correspond to the inlet Reynolds numbers of 100, 250, 500, 750, and 1000 and is listed in Table 1.

### 2.5. Streamlines and Masslines Visualization

The concept of masslines evolved from the use of streamfunction and streamlines to visualize fluid flow. In cartesian coordinates, the stream function is defined as

$$\frac{\partial \psi}{\partial y} = u, \ -\frac{\partial \psi}{\partial x} = v \tag{30}$$

where $\psi(x, y)$ is the streamfunction. The flow is locally parallel to the constant line of the streamfunction (streamlines). Thus, although there is no explicit substitution for the velocity component $(u, v)$ as the

source of the local flow attributes, constant streamlines provide a valuable observation of the fluid flow and its characteristics. Similarly, mass-function and masslines are introduced as a visualization aid of the mass transfer by convection and diffusion mechanism, described as

$$\frac{\partial M}{\partial y} = M_x, \ -\frac{\partial M}{\partial y} = M_y \tag{31}$$

where

$$M_x = \rho u \omega_i - \rho D_i \frac{\partial \omega_i}{\partial x}, \ M_y = \rho v \omega_i - \rho D_i \frac{\partial \omega_i}{\partial y} \tag{32}$$

*2.6. Numerical Methodology*

The ANSYS Design modeler and ANSYS Meshing were used to create, mesh, and label the computational domain, consisting of a straight reactor and curved reactor. To study the dependency of the numerical result on the amount of generated mesh, several mesh sizes were prepared. The computational domain was then exported to ANSYS Fluent for model set-up by incorporating the developed conservation equations together with constitutive relations and corresponding boundary conditions. The widely adopted Semi-Implicit-Pressure-Linked equation (SIMPLE) algorithm, second order upwind discretization, and algebraic multi-grid (AMG) method were chosen to solve the developed model. A residual criterion of $10^{-6}$ was set for all parameters. The computational model takes approximately 1–2 h to reach convergence by using a single processor setting in a high-performance computer (HPC). A range of 5 GB to 10 GB RAM utilization was recorded during the computational run where other processes were closed.

To evaluate the mesh independency of the numerical result, mesh independent studies were conducted using the previously prepared mesh. The results are presented in Table 4. As can be seen, no significant changes were observed for mesh beyond 994,000. To obtain firm confirmation, the spatial flow profile and methane concentration are presented in Figures 2 and 3. From these results, a mesh size amounting to 994,000 mesh was chosen for the remaining cases.

**Table 4.** Example of a mesh independent test for square cross-section. The mesh used is highlighted in bold.

| Total Mesh | Straight Tube | | Helical Coil Tube | |
|---|---|---|---|---|
| | Pressure Drop (Pa) | Methane Consumption | Pressure Drop (Pa) | Methane Consumption |
| 80,000 | 5190.42 | 0.40 | 11,682.55 | 0.44 |
| 160,000 | 5173.41 | 0.40 | 11,682.55 | 0.44 |
| 360,000 | 5221.76 | 0.40 | 11,636.86 | 0.44 |
| 720,000 | 5221.21 | 0.40 | 11,594.56 | 0.44 |
| **994,000** | **5221.36** | **0.40** | **11,543.44** | **0.44** |
| 1,280,000 | 5220.97 | 0.40 | 11,543.23 | 0.44 |
| 1,740,000 | 5220.56 | 0.40 | 11,542.98 | 0.44 |

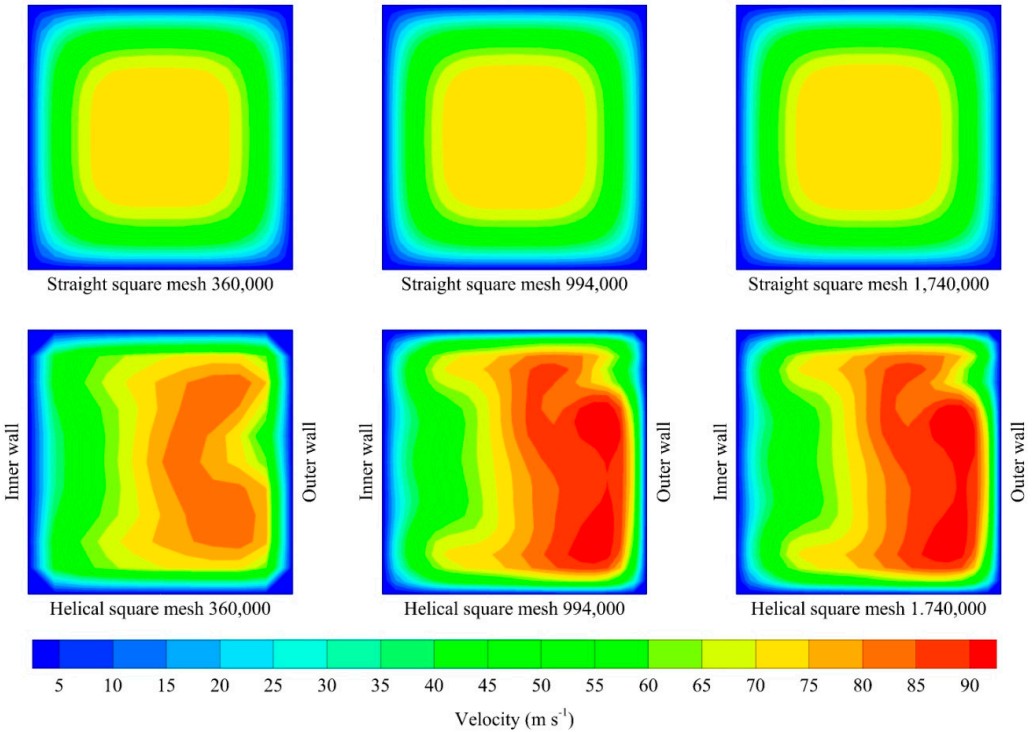

**Figure 2.** Axial velocity profile (contour, m s$^{-1}$) of flow in straight and helical squares for various mesh.

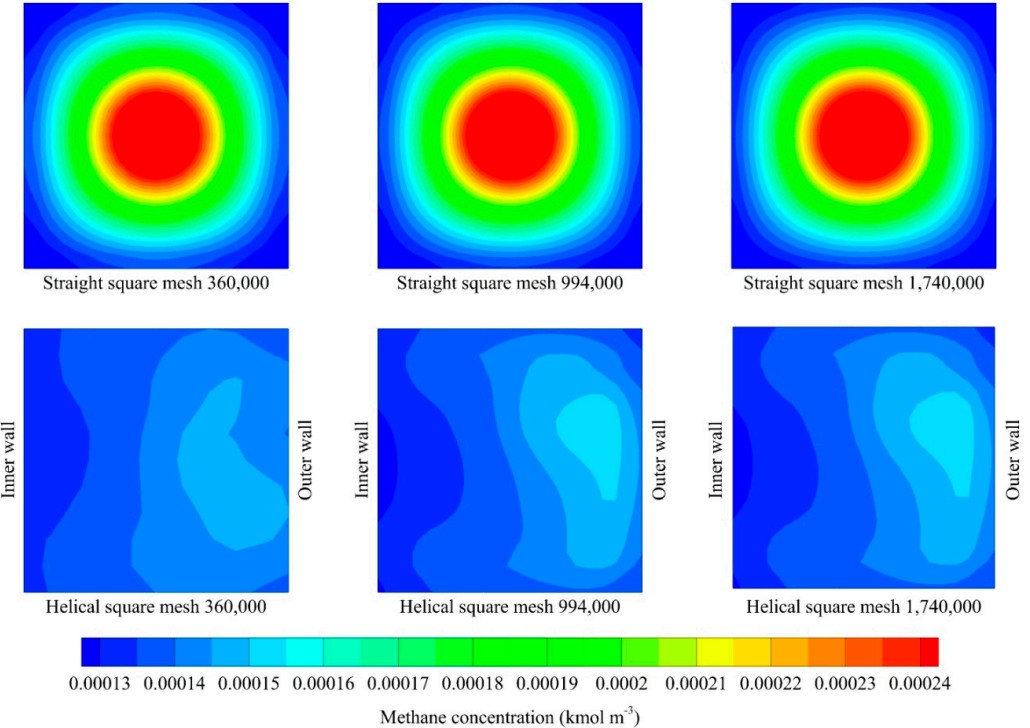

**Figure 3.** Methane concentration (kmol m$^{-3}$) of flow in straight and helical squares for various mesh.

## 3. Results and Discussion

### 3.1. Model Validation

Validation of the developed model against the experimental counterpart and/or other numerical study was conducted to assess the validity and accuracy of the model. The results of the presented

model were compared to the experimental data by Bond et al. [40] and results of the numerical studies
by Bond et al. [40] and Canu [41]. The methane conversion and the temperature distribution in a
monolithic reactor were estimated using single channel flow approximation. The reactor was an 8-cm
long, 1.5-mm × 1.5-mm square channel with an inlet velocity of 7.3 m/s. The catalytic oxidation reaction
starts with a low stoichiometric gas inlet ($\xi = 0.18$) and subsequently a higher inlet stoichiometry
($\xi = 0.39$) was implemented. A more detailed description of the reaction kinetics can be found in
Canu [41] and Bond et al. [40]. In terms of methane conversion, the model demonstrated good
agreement with the experimental results, as shown in Figure 4a. It can be observed that the model
gave better prediction than the models developed by Bond et al. [40] and Canu [41], especially at high
methane stoichiometry. The model was also able to predict temperature distribution in the reactor
sufficiently at high and low stoichiometry, as depicted in Figure 4b.

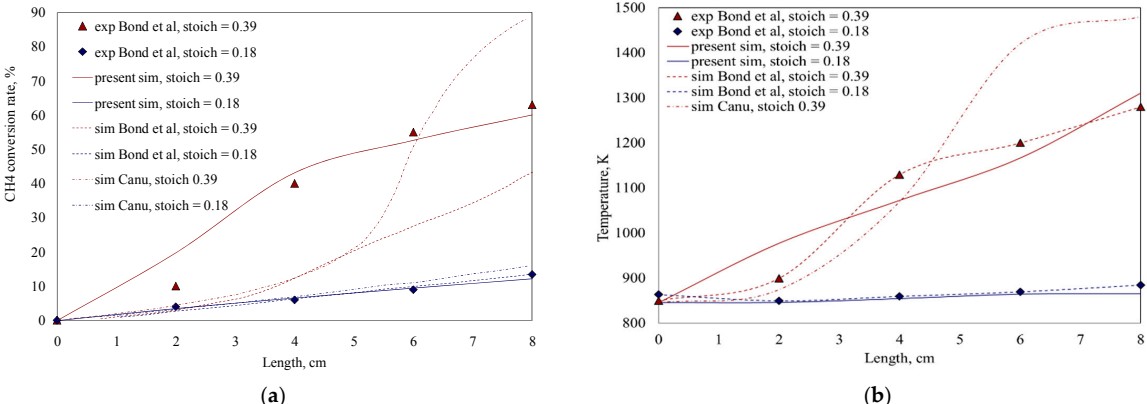

**Figure 4.** Experimental data by Bond et al. [40] and comparison with numerical data by
Bond et al. [40] and Canu [41] at low and high inlet stoichiometry for (**a**) methane conversion and
(**b**) temperature distributions.

### 3.2. Effect of Reactor Geometry

Two reactor channel geometries were considered in this study: straight and helical channels.
Figure 5 illustrates the axial airflow velocity profile in straight and helical channels for three different
cross-sectional geometries (circle, ellipse, and square). The presence of secondary flow can be observed
in the helical channels, while the flow in straight channels exhibits a fully developed behavior.
The velocity profile in all helical channels shows that the axial airflow velocity increases outwards,
with the highest velocity near the outer wall due to the presence of centrifugal force from the curvature
geometry which leads to significant radial pressure gradients in flow core region. In proximity of the
inner and outer walls, however, the axial velocity and the centrifugal force will approach zero. Hence,
to balance the momentum transport, secondary flow should develop along the outer wall. In addition,
vortices development was observed in the helical channels as a result of the presence of secondary
flow. Considering the methane concentration distribution, Figure 6 demonstrates higher methane
concentration in straight channels, by approximately 60%, compared to helical channels, suggesting a
smaller methane reaction. Methane concentration was highest at the center of the straight channels
due to the fully developed behavior of the flow, and lowest at the wall of straight channels, owing to
catalytic surface reaction taking place at the wall. In the helical channels, the high axial velocity near
the outer wall was mirrored by high methane concentration. On the other hand, the concentration of
methane was lower where more vortices were present. This implies that the vortex of the secondary
flow enhanced mass transfer, and hence, the reaction. Figure 7 shows methane conversion along both
channel geometries at various Reynolds number. In general, methane conversion along channel length
was higher for helical channels. Methane conversion follows an asymptotic behavior which levels off
faster at a low Re number, becoming more linear as the inlet Re number increases. It can also be seen

that increasing inlet Re number by 5, 10, and 15 times decreases the conversion by 30%, 55%, and 65%, respectively, as a consequence of decreasing methane residence time.

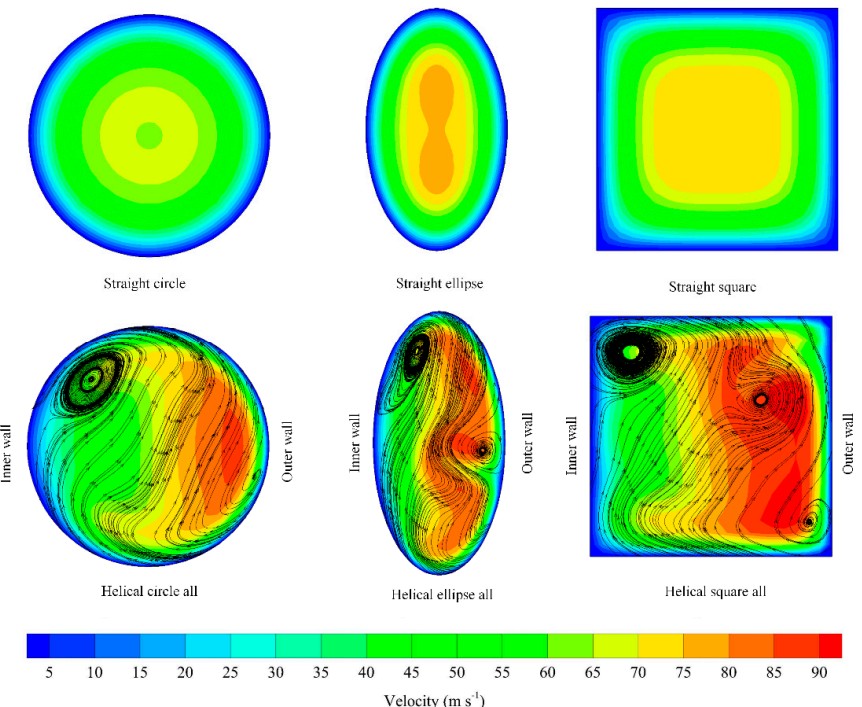

**Figure 5.** Axial velocity profile (contour, m s$^{-1}$) and streamlines of flow in (**a**) straight circle, (**b**) straight ellipse, (**c**) straight square, (**d**) helical circle, (**e**) helical ellipse, and (**f**) helical square tubes at L = 31 mm and Reynolds number of 1000.

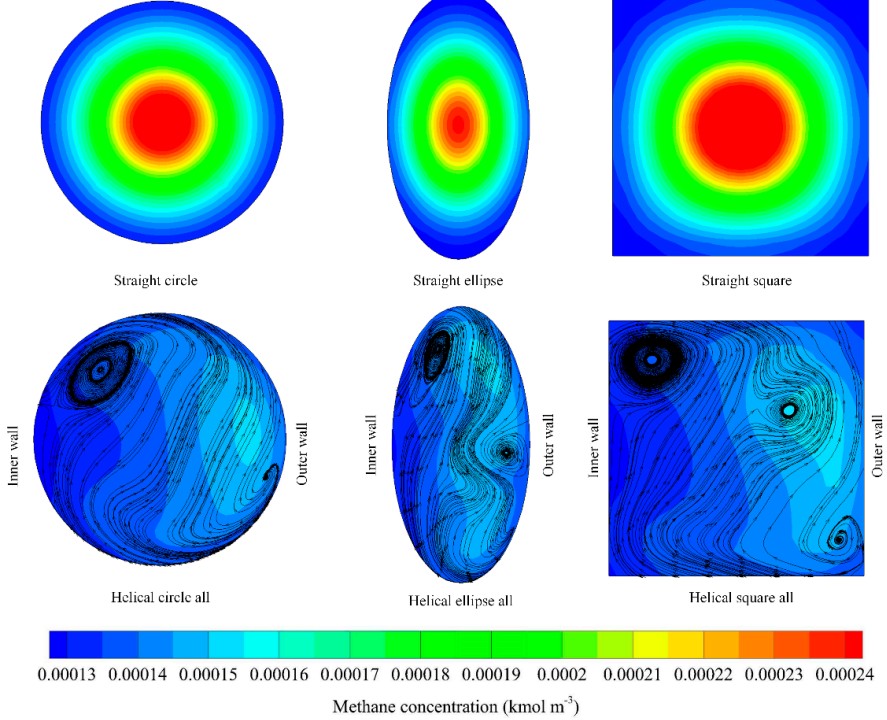

**Figure 6.** Methane concentration (kmol m$^{-3}$) and masslines in (**a**) straight circle, (**b**) straight ellipse, (**c**) straight square, (**d**) helical circle, (**e**) helical ellipse, and (**f**) helical square tubes at L = 31 mm and Reynolds number of 1000.

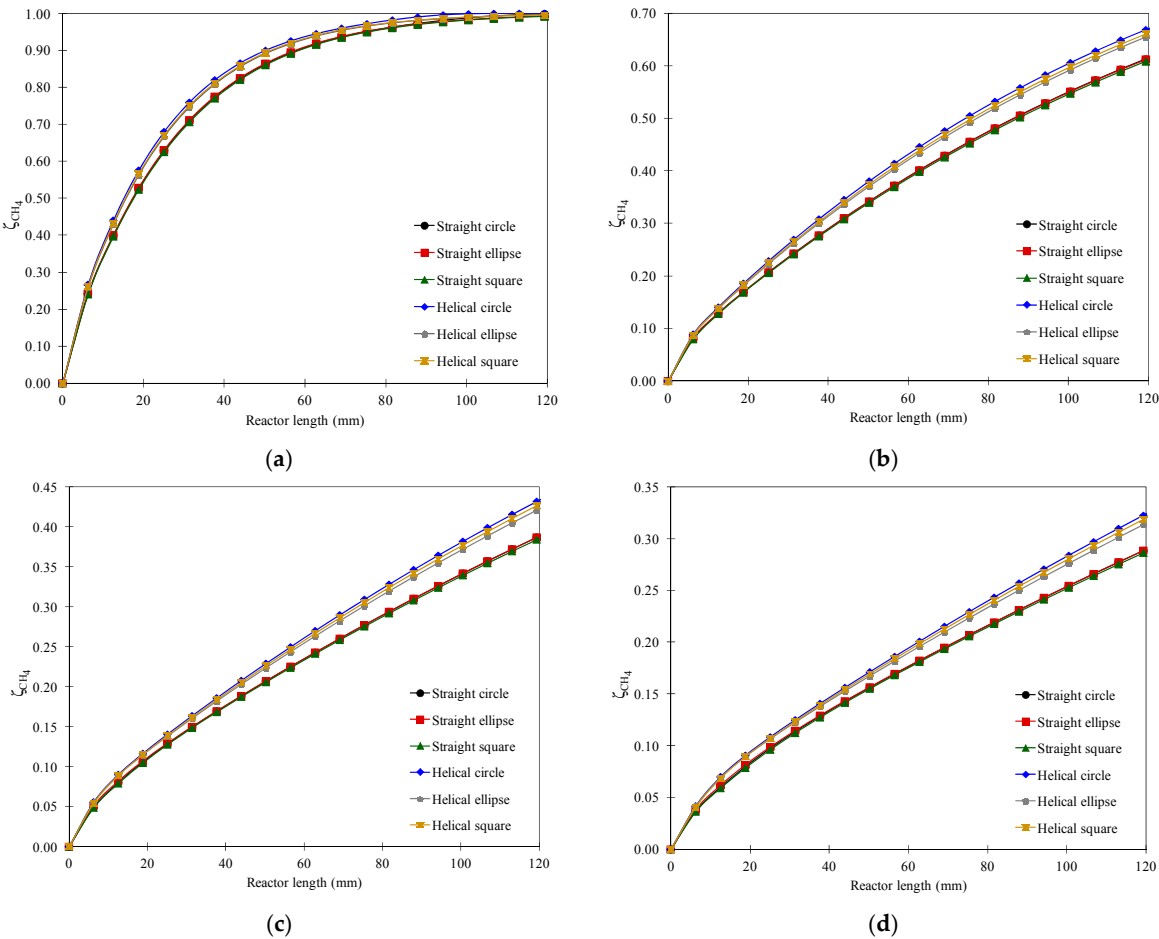

**Figure 7.** Methane conversion along the straight and helical reactor at Reynolds numbers of (**a**) 100, (**b**) 500, (**c**) 1000, and (**d**) 1500.

### 3.3. Effect of Cross-Section Area

The effect of the cross-sectional area was also investigated by modelling various geometries, i.e., circle, ellipse, and square, while keeping the hydraulic diameter constant. As illustrated in Figures 5 and 6, all three geometries show virtually identical velocity and methane concentration profiles. The only notable difference is in the number of vortex flow present: 1, 2, and 3 for circle, ellipse, and square, respectively; for all cross-sections, the vortex appeared at the top left of the inner wall, an additional vortex at the mid of outer wall developed in an ellipse and square cross-sections, and the third vortex generated at the bottom of outer wall in a square cross-section. When comparing conversions, Figure 7 shows that the helical circle channel gave the best performance of all the Reynolds numbers, with a methane conversion approximately 5% and 3% higher than that of the helical ellipse and helical square, respectively.

### 3.4. Effect of Selective Coating Strategies

It is well known that catalysts are usually made from noble, expensive metals, such as Platinum. A reduction in catalyst coating is expected to reduce the cost of the reactor. The objective of the study was to investigate potential catalyst savings through selective coating. Different catalyst coating strategies were examined: inner wall coating, outer wall coating, and all wall coating, and all three were implemented to the circle, ellipse, and square geometries. The effect of different coating configurations on the axial airflow velocity profile is shown in Figure 8.

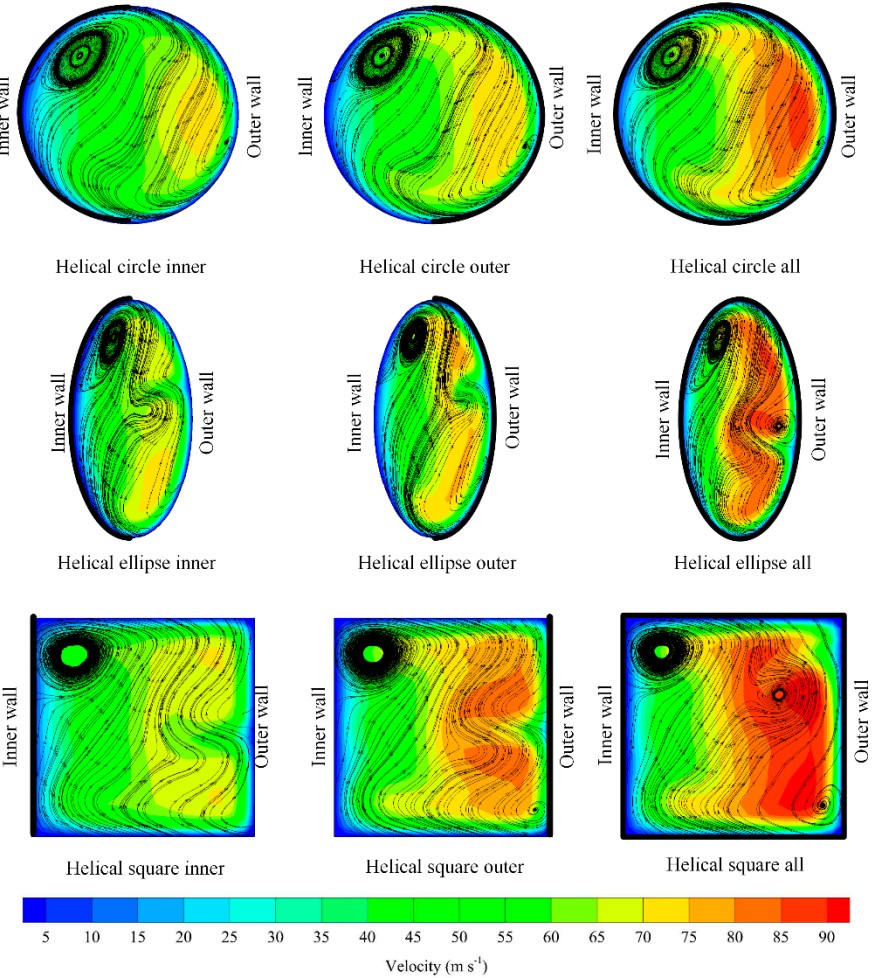

**Figure 8.** Axial velocity profile (contour, m s$^{-1}$) and streamlines of flow in helical reactors with various cross-sections and coating strategies at L = 31 cm and Reynolds number of 1000. The coated walls are highlighted in black line.

While all three coatings display similar velocity profiles, closer inspection reveals that all wall coating leads to the highest axial velocity for each geometry which can be attributed to the additional momentum due to surface reactions in all walls. In the helical circle channel, the all wall coating gave the highest velocity value of 90 m/s while both inner and outer coatings yield maximum velocity of 70 m/s. For the helical ellipse, the axial velocity observed in the all wall coating configuration was 6.25% and 13.33% higher than that observed in the outer and inner configurations, respectively. The same trend can be seen in helical square channel, with an even more noticeable difference; velocity in the all wall coating being 12.5% greater as compared to outer coating and 28.6% greater than inner coating. It can also be observed from Figure 8 that more vortices are present when the catalyst is coated onto all of the walls. Calculating the swirl number which is defined as the ratio of axial flux of the tangential momentum to the axial flux of the axial momentum [42,43], it was found that, for Re 1000, the helical ellipse with the inner coating had the highest swirl number of $2.25 \times 10^{-4}$ followed by the same geometry with the outer and all coating which were $1.47 \times 10^{-4}$ and $1.44 \times 10^{-4}$, respectively. This indicates a stronger secondary flow for the helical reactor with an ellipse cross-section. Meanwhile, the lowest swirl number was found for the helical circle with the outer coating which had a swirl number of $3.04 \times 10^{-4}$, followed by the helical circle with the all coating at $3.09 \times 10^{-4}$, and the helical with the circle inner coating at $3.95 \times 10^{-4}$.

The distribution of methane concentration in Figure 9 illustrates that the all wall coating generated the highest methane conversion, indicated by the lower methane concentration, followed by the outer

and inner wall coatings. Additionally, the profile of the masslines showed strong similarity to the streamlines profile, from which it can be concluded that the mass transfer was convection dominated. The study also investigated methane conversion along the helical reactor for all cross-sectional geometry and coating configuration combinations at various Reynolds numbers, as presented in Figure 10. In general, the all wall coating resulted in a higher methane conversion for all cross-sectional geometries, followed by the outer and inner coatings. At a low Reynolds number (Re = 100), the all wall coating gave up to 98% conversion, whereas the outer and inner coatings give roughly 93% and 89%, respectively. At a high Reynolds number (Re = 1500), up to 32% conversion is was achieved with the all wall coating, followed by the outer coating with 19% conversion and the inner coating with 17% conversion. The observed trend indicates that the performance of the all wall coating configuration was more significant when compared to the inner and outer coatings at high Re numbers. Looking into the cross-sectional performances, Figure 10 illustrates that overall, the circle cross-sectional geometry performed best, followed by the square and ellipse, although the difference was marginal.

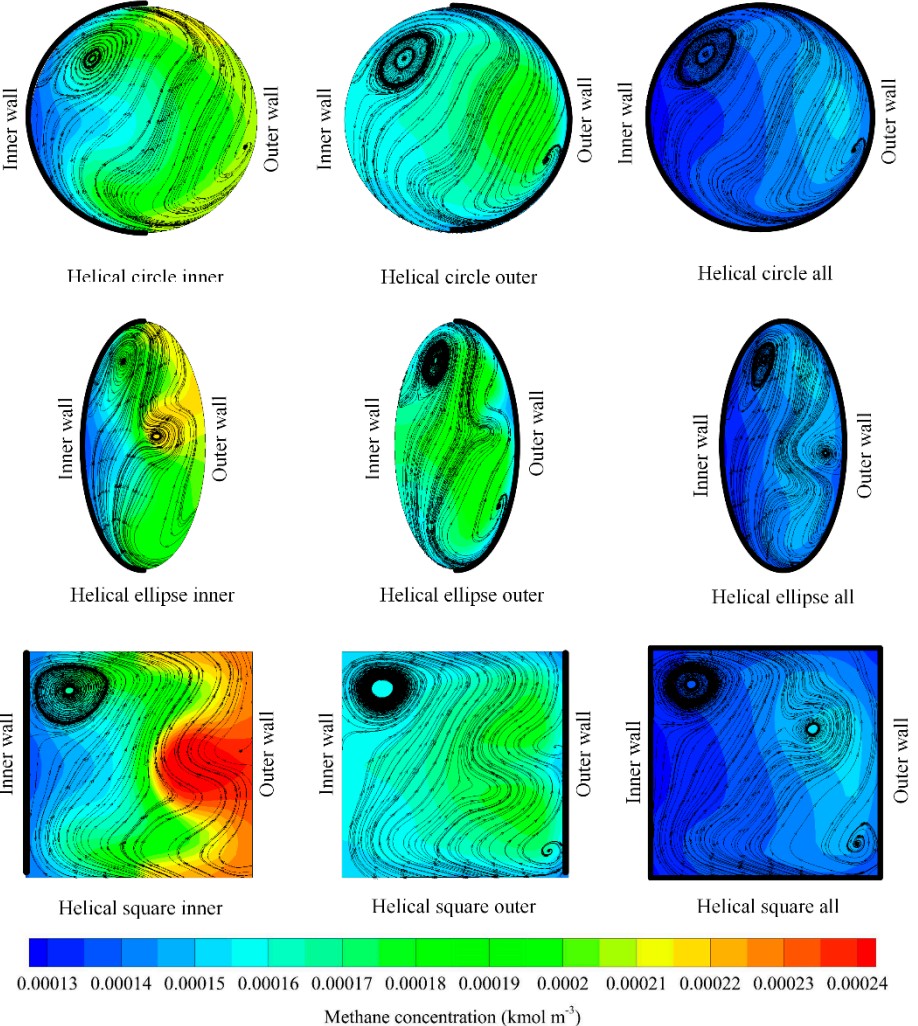

**Figure 9.** Methane concentration (kmol m$^{-3}$) and masslines in the helical reactors with various cross-sections and coating strategies at L = 31 cm and Reynolds number of 1000. The coated walls are highlighted in black line.

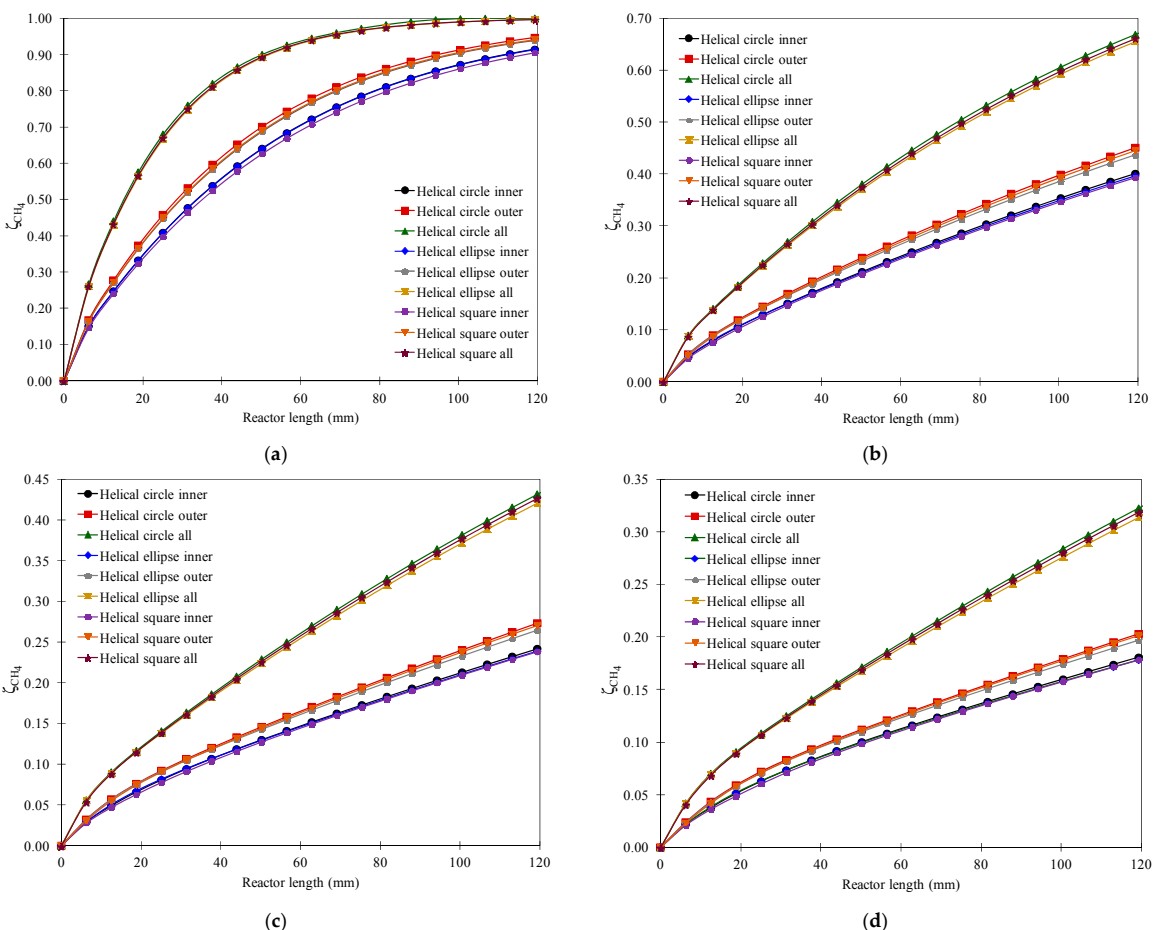

**Figure 10.** Methane conversion along the helical reactors at Reynolds numbers of (**a**) 100, (**b**) 500, (**c**) 1000, and (**d**) 1500.

### 3.5. Effect of Channel Length

Another important factor that must be taken into account in designing a reactor is the channel length in order to optimize the performance and cost. Table 5 shows the power generated by methane combustion for different channel lengths: 31, 63, 94, and 126 mm in Watt per channel. Note that in practical application, the reactor may comprise hundreds or thousands of channels stacked up in monolith. Here. it is evident that increasing the channel length increases the methane-to-energy conversion. On average, the amount of energy generated in straight reactors (circle, ellipse, square) at Re = 100 increases by 29%, 38%, and 41% when the length is increased by two, three, and four times, respectively. These increases become 64%, 117%, 159% at Re = 500; 62%, 117%, 167% at Re = 1000; and 60%, 113%, 163% at Re = 1500. For the helical circle design, average increases in energy generation when the channel length is doubled, tripled, and quadrupled (from base length of 31 mm) are, respectively, 41%, 62%, and 71% for Re = 100; 65%, 121%, and 169% for Re = 500; 62%, 118%, and 169% for Re = 1000; and 58%, 111%, and 160% for Re = 1500. Overall, increasing the channel length at all Re number increases energy generation significantly, although the increase was not linear. Furthermore, enhancement in energy generation by helical all-wall-coated reactors compared to straight reactors was more significant at high Re numbers.

It is obvious that increasing the length of reactor channels increases flow resistance and parasitic load (pumping power). Table 6 tabulates the pumping power required per channel in mW. In straight channels, the increase in pressure drop was in linear relation with the Reynolds number and length which follows Moody's chart for friction factor correlation. In the helical channel, on the other hand, the pumping power increases exponentially with the Reynolds number; notably the pumping power

increased by approximately 600 times as the Reynolds increased from 100 to 1500. As the channel length increased, the pumping power increased proportionally following the Darcy–Weisbach equation. Amongst all cross-sections investigated, the square channel gave rise to the highest pumping power followed by the ellipse (about 10 to 20% lower) and circle (about 20–40% lower). It is also worth mentioning that selective coating marginally affects pumping power requirements. The all walls coating yielded the highest pumping power, followed by the outer and inner walls. This was due to the temperature- and species-dependent thermophysical properties used in our model.

**Table 5.** Power generation from methane combustion in single channel ventilation air methane (VAM) reactor (Watt).

| Geometry | Reactor 31 mm | | | | Reactor 63 mm | | | |
|---|---|---|---|---|---|---|---|---|
| | **Re 100** | **Re 500** | **Re 1000** | **Re 1500** | **Re 100** | **Re 500** | **Re 1000** | **Re 1500** |
| Straight circle | 0.21 | 0.36 | 0.45 | 0.51 | 0.27 | 0.60 | 0.73 | 0.82 |
| Straight ellipse | 0.23 | 0.40 | 0.49 | 0.56 | 0.30 | 0.65 | 0.79 | 0.89 |
| Straight square | 0.24 | 0.41 | 0.50 | 0.57 | 0.31 | 0.67 | 0.81 | 0.92 |
| Helical circle inner | 0.14 | 0.23 | 0.28 | 0.33 | 0.22 | 0.37 | 0.45 | 0.52 |
| Helical circle outer | 0.16 | 0.25 | 0.32 | 0.37 | 0.23 | 0.42 | 0.51 | 0.58 |
| Helical circle all | 0.23 | 0.40 | 0.49 | 0.56 | 0.28 | 0.67 | 0.81 | 0.90 |
| Helical ellipse inner | 0.16 | 0.24 | 0.31 | 0.36 | 0.24 | 0.40 | 0.49 | 0.56 |
| Helical ellipse outer | 0.17 | 0.27 | 0.34 | 0.40 | 0.25 | 0.45 | 0.54 | 0.62 |
| Helical ellipse all | 0.24 | 0.43 | 0.52 | 0.60 | 0.31 | 0.71 | 0.86 | 0.96 |
| Helical square inner | 0.16 | 0.25 | 0.31 | 0.36 | 0.24 | 0.41 | 0.50 | 0.58 |
| Helical square outer | 0.18 | 0.28 | 0.35 | 0.41 | 0.26 | 0.47 | 0.57 | 0.65 |
| Helical square all | 0.25 | 0.45 | 0.55 | 0.63 | 0.32 | 0.74 | 0.90 | 1.01 |
| Geometry | Reactor 94 mm | | | | Reactor 126 mm | | | |
| | **Re 100** | **Re 500** | **Re 1000** | **Re 1500** | **Re 100** | **Re 500** | **Re 1000** | **Re 1500** |
| Straight circle | 0.29 | 0.79 | 0.97 | 1.09 | 0.30 | 0.94 | 1.20 | 1.34 |
| Straight ellipse | 0.32 | 0.86 | 1.06 | 1.19 | 0.32 | 1.03 | 1.31 | 1.47 |
| Straight square | 0.33 | 0.89 | 1.09 | 1.22 | 0.34 | 1.06 | 1.34 | 1.51 |
| Helical circle inner | 0.26 | 0.50 | 0.61 | 0.69 | 0.28 | 0.62 | 0.75 | 0.84 |
| Helical circle outer | 0.27 | 0.57 | 0.68 | 0.77 | 0.29 | 0.70 | 0.85 | 0.95 |
| Helical circle all | 0.30 | 0.87 | 1.09 | 1.21 | 0.30 | 1.03 | 1.34 | 1.50 |
| Helical ellipse inner | 0.28 | 0.54 | 0.65 | 0.74 | 0.30 | 0.67 | 0.81 | 0.91 |
| Helical ellipse outer | 0.29 | 0.60 | 0.72 | 0.81 | 0.31 | 0.74 | 0.90 | 1.00 |
| Helical ellipse all | 0.32 | 0.93 | 1.16 | 1.29 | 0.33 | 1.10 | 1.42 | 1.59 |
| Helical square inner | 0.28 | 0.56 | 0.67 | 0.76 | 0.31 | 0.69 | 0.83 | 0.94 |
| Helical square outer | 0.30 | 0.63 | 0.77 | 0.86 | 0.32 | 0.78 | 0.95 | 1.06 |
| Helical square all | 0.33 | 0.97 | 1.21 | 1.35 | 0.34 | 1.15 | 1.49 | 1.68 |

**Table 6.** Required pumping power for single channel reactor VAM reactor (mWatt).

| Geometry | Reactor 31 mm | | | | Reactor 63 mm | | | |
|---|---|---|---|---|---|---|---|---|
| | **Re 100** | **Re 500** | **Re 1000** | **Re 1500** | **Re 100** | **Re 500** | **Re 1000** | **Re 1500** |
| Straight circle | 0.15 | 5.08 | 23.78 | 58.57 | 0.29 | 8.66 | 40.33 | 99.47 |
| Straight ellipse | 0.22 | 7.25 | 33.91 | 83.54 | 0.42 | 12.50 | 57.64 | 141.99 |
| Straight square | 0.22 | 7.36 | 34.34 | 84.60 | 0.41 | 12.58 | 58.54 | 144.12 |
| Helical circle inner | 0.18 | 6.34 | 32.08 | 83.92 | 0.36 | 13.40 | 68.86 | 181.16 |
| Helical circle outer | 0.18 | 6.78 | 34.73 | 91.99 | 0.36 | 13.92 | 72.26 | 191.86 |
| Helical circle all | 0.20 | 7.81 | 41.16 | 109.98 | 0.38 | 14.99 | 79.11 | 211.62 |
| Helical ellipse inner | 0.24 | 8.06 | 39.89 | 100.87 | 0.48 | 16.74 | 83.80 | 215.78 |
| Helical ellipse outer | 0.24 | 8.49 | 41.80 | 110.28 | 0.49 | 17.24 | 87.34 | 227.95 |
| Helical ellipse all | 0.26 | 9.86 | 51.53 | 137.52 | 0.51 | 18.65 | 98.68 | 263.26 |
| Helical square inner | 0.25 | 8.96 | 45.26 | 118.11 | 0.51 | 19.37 | 97.98 | 260.38 |
| Helical square outer | 0.25 | 9.89 | 52.12 | 137.39 | 0.52 | 20.23 | 108.93 | 285.22 |
| Helical square all | 0.28 | 11.59 | 61.47 | 162.61 | 0.54 | 22.03 | 116.56 | 313.94 |

**Table 6.** *Cont.*

| Geometry | Reactor 94 mm | | | | Reactor 126 mm | | | |
|---|---|---|---|---|---|---|---|---|
| | **Re 100** | **Re 500** | **Re 1000** | **Re 1500** | **Re 100** | **Re 500** | **Re 1000** | **Re 1500** |
| Straight circle | 0.42 | 12.04 | 55.03 | 135.63 | 0.55 | 15.39 | 68.97 | 169.12 |
| Straight ellipse | 0.62 | 17.52 | 79.11 | 194.03 | 0.82 | 22.51 | 99.68 | 242.81 |
| Straight square | 0.60 | 17.46 | 80.03 | 197.06 | 0.80 | 22.29 | 100.28 | 246.04 |
| Helical circle inner | 0.54 | 20.56 | 106.49 | 281.14 | 0.73 | 27.85 | 145.34 | 385.47 |
| Helical circle outer | 0.55 | 21.09 | 110.07 | 292.66 | 0.73 | 28.39 | 148.96 | 397.24 |
| Helical circle all | 0.56 | 22.16 | 116.95 | 312.60 | 0.74 | 29.46 | 155.84 | 417.20 |
| Helical ellipse inner | 0.72 | 25.52 | 128.72 | 337.60 | 0.97 | 34.41 | 174.88 | 461.52 |
| Helical ellipse outer | 0.73 | 26.02 | 132.44 | 349.02 | 0.97 | 34.92 | 178.65 | 475.26 |
| Helical ellipse all | 0.75 | 27.44 | 145.64 | 388.09 | 0.99 | 36.34 | 193.58 | 515.40 |
| Helical square inner | 0.77 | 29.84 | 151.74 | 408.53 | 1.04 | 40.35 | 207.47 | 563.96 |
| Helical square outer | 0.78 | 30.64 | 163.31 | 434.95 | 1.04 | 41.22 | 219.15 | 590.81 |
| Helical square all | 0.81 | 32.48 | 170.90 | 463.43 | 1.07 | 43.10 | 226.72 | 619.15 |

## 3.6. Overall Performance

The overall performance of the reactors was examined using the figure of merit (FoM) concept, defined as net power generation (power generated–pumping power) per unit mass of catalyst (Watt per mg of Pt), as can be inferred from Table 7. Here several features were apparent; foremost among them was the shorter channel and higher Reynolds number yield higher than FoM. Notably, when the channel length increased by four times, the FoM decreased by about 50%, while increasing the Reynolds from 100 to 1500 increased the FoM by approximately two times. The helical channels outperformed the straight channel, especially at shorter channels and lower Reynolds numbers. Looking at the channel cross-sections, the circular cross-section performed the best FoM, followed by the ellipse and square by about 10% and 15%, respectively. It is also worth mentioning that the inner coating yielded the best FoM especially at a lower Reynolds numbers, followed by the outer coating and all coating by 10% and 20%, respectively. Closer inspection reveals that at a high Reynolds number and long channel, the straight channels gave rise to a higher FoM. Therefore, it can be deduced that a selective inner wall coating is best when applied for short channels at a high Reynolds number.

**Table 7.** Figure of Merit (FoM) for a single channel VAM reactor (Watt per mg of Pt).

| Geometry | Reactor 31 mm | | | | Reactor 63 mm | | | |
|---|---|---|---|---|---|---|---|---|
| | **Re 100** | **Re 500** | **Re 1000** | **Re 1500** | **Re 100** | **Re 500** | **Re 1000** | **Re 1500** |
| Straight circle | 3.60 | 6.08 | 7.19 | 7.66 | 2.33 | 5.02 | 5.82 | 6.09 |
| Straight ellipse | 3.61 | 6.06 | 7.09 | 7.44 | 2.33 | 5.00 | 5.72 | 5.84 |
| Straight square | 3.59 | 6.03 | 7.04 | 7.30 | 2.33 | 4.97 | 5.70 | 5.82 |
| Helical circle inner | 5.31 | 8.16 | 9.29 | 9.11 | 4.01 | 6.70 | 7.17 | 6.32 |
| Helical circle outer | 4.94 | 7.68 | 8.83 | 8.75 | 3.62 | 6.35 | 6.81 | 6.06 |
| Helical circle all | 3.85 | 6.69 | 7.59 | 7.63 | 2.39 | 5.52 | 6.17 | 5.84 |
| Helical ellipse inner | 5.20 | 7.92 | 8.99 | 8.63 | 3.93 | 6.47 | 6.82 | 5.83 |
| Helical ellipse outer | 4.93 | 7.60 | 8.74 | 8.43 | 3.63 | 6.23 | 6.61 | 5.69 |
| Helical ellipse all | 3.79 | 6.51 | 7.33 | 7.20 | 2.38 | 5.37 | 5.91 | 5.41 |
| Helical square inner | 5.20 | 7.93 | 8.68 | 8.01 | 3.96 | 6.52 | 6.72 | 5.28 |
| Helical square outer | 4.84 | 7.47 | 8.31 | 7.64 | 3.57 | 6.18 | 6.34 | 5.02 |
| Helical square all | 3.81 | 6.57 | 7.29 | 6.98 | 2.38 | 5.42 | 5.90 | 5.21 |
| Helical circle all | 1.68 | 4.79 | 5.49 | 5.09 | 1.26 | 4.23 | 5.01 | 4.59 |

**Table 7.** *Cont.*

| Geometry | Reactor 94 mm | | | | Reactor 126 mm | | | |
|---|---|---|---|---|---|---|---|---|
| | Re 100 | Re 500 | Re 1000 | Re 1500 | Re 100 | Re 500 | Re 1000 | Re 1500 |
| Straight circle | 1.66 | 4.41 | 5.21 | 5.40 | 1.27 | 3.94 | 4.80 | 4.98 |
| Straight ellipse | 1.65 | 4.39 | 5.11 | 5.17 | 1.26 | 3.92 | 4.71 | 4.76 |
| Straight square | 1.65 | 4.37 | 5.09 | 5.15 | 1.26 | 3.90 | 4.69 | 4.74 |
| Helical circle inner | 3.16 | 6.01 | 6.22 | 5.02 | 2.57 | 5.53 | 5.64 | 4.23 |
| Helical circle outer | 2.78 | 5.69 | 5.97 | 4.93 | 2.22 | 5.21 | 5.45 | 4.26 |
| Helical circle all | 1.68 | 4.79 | 5.49 | 5.09 | 1.26 | 4.23 | 5.01 | 4.59 |
| Helical ellipse inner | 3.10 | 5.79 | 5.87 | 4.49 | 2.52 | 5.33 | 5.31 | 3.72 |
| Helical ellipse outer | 2.81 | 5.58 | 5.74 | 4.51 | 2.24 | 5.11 | 5.22 | 3.81 |
| Helical ellipse all | 1.66 | 4.67 | 5.25 | 4.67 | 1.26 | 4.13 | 4.78 | 4.20 |
| Helical square inner | 3.14 | 5.84 | 5.78 | 3.93 | 2.56 | 5.38 | 5.20 | 3.09 |
| Helical square outer | 2.75 | 5.53 | 5.52 | 3.89 | 2.20 | 5.07 | 5.02 | 3.21 |
| Helical square all | 1.67 | 4.71 | 5.24 | 4.47 | 1.26 | 4.16 | 4.76 | 3.98 |

## 4. Conclusions

A numerical investigation was conducted to evaluate reaction performance and potential catalyst savings of a helical coil reactor. Several coating strategies were implemented and evaluated, i.e., inner wall coating, outer wall coating, and all wall coating. The results revealed several important findings: (1) the helical channel reactors performed better compared to the straight channel reactors; (2) the presence of secondary flow enhanced the convective mixing and mass transport; (3) the channel with the circular cross-section yielded better a reaction compared to the ellipse and square counterparts; (4) a shorter channel and a higher Reynolds number was beneficial to getting a higher figure of merit (net power generated per mg of Pt catalyst). Future work will focus on optimization of the reactor design for cost-effective ventilation air methane energy extraction.

**Author Contributions:** Conceptualization, A.P.S. and J.C.K.; methodology, J.C.K. and A.P.S.; software, J.C.K., D.C.L., B.A.C., and A.P.S.; validation, J.C.K. and A.P.S.; formal analysis, J.C.K., B.A.C., and A.P.S.; investigation, J.C.K., B.A.C., and A.P.S.; resources, J.C.K. and A.P.S.; data curation, J.C.K., D.C.L., B.A.C., and A.P.S.; writing—original draft preparation, J.C.K., B.A.C., L.C., L.J., and A.P.S.; writing—review and editing, A.P.S. and J.C.K.; visualization, J.C.K. and B.A.C.; supervision, J.C.K. and A.P.S.; project administration, A.P.S., L.C., and L.J.; funding acquisition, A.P.S., L.C., L.J.

**Funding:** This research and the APC was funded by Shandong University of Science and Technology through SDUST Open Grant.

**Acknowledgments:** The first author gratefully acknowledges the facility and financial support from the Yayasan Universiti Teknologi PETRONAS (YUTP) through YUTP Fundamental Research Grant (YUTP-FRG) no 0153AA-E64. The second, fourth, fifth, and sixth authors gratefully acknowledge the financial support from Shandong University of Science and Technology through Open Grant.

**Conflicts of Interest:** The authors declare no conflict of interest.

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
