# Peer review of "Numerical Evaluation of Potential Catalyst Savings for Ventilation Air Methane Catalytic Combustion in Helical Coil Reactors with Selective Wall Coating"

_catalysts, doi:10.3390/catal9040380_

Round 1
Reviewer 1 Report
The paper is worthwhile and merits acceptance after addressing several matters below:
Major:
1. The reactor wall temperature was fixed at 1290 K. Why such temperature and what is the meaning to externally heat a source of heat to maintain such wall temperature? The external temperature shall be determined by the heating power and heat loses to the environment.
Minor:
1. The introduction is adequate. However, it may benefit from a broader overview of catalyst-coated reactors to give a broader context. Examples of the works include DOI: 10.3390/catal8020058, 10.1016/j.cattod.2017.09.041, 10.1016/j.cej.2017.06.161, 10.3390/catal7120358
2. Section 2 seems rather too technical for a general reader. It may be worthwhile to summarise the section in a paragraph or two and put the original section into the appendix or supplementary.
3. Table 1 provides a dazzling array of various parameters which may benefit justification. Do the coils have the same length as the straight sections? Why are sizes (areas) of various channels different? Why platinum coverage is measured in kg*mol/m^2, not in kg/m^2 or mol/m^2
4. Abbreviations and units in all the tables require clarification. Tables 2,4 in particular
5. Table 3 seem to present too minor details and does not warrant presence in the main text
6. The caption of table 4 does not explain what is the cross-section shape of the tube used
7. Figure 2. Simulations from the literature that do not fit the experiment only obstruct the view and does not contribute. It may be better to leave current simulations and experimental points only.
8. Figure 4 and later. Please check units of concentration – these are not kmol, and neither mol. It must be something like mol/m^3
9. Figure 5 and later. Why conversion is called “conversion rate”? The term “rate” implies the speed of change
10. Figure 6. Coated sections are written under the tube. It may be clearer to draw them on the outside of the figures to make it easier to understand
11. It may be good to do a minor proof-read and eliminate a few typos and eliminate or explain jargon such as “gassy mines”
Author Response
We are very grateful for the positive reviews, comments, suggestions and the time spent reviewing our paper (Ref. No.: catalysts-481258). In the following, we have addressed all the issues raised by the reviewer in the revised paper. Below, we list the reviewer’s comments followed by our responses. The changes are highlighted in yellow in the revised manuscript:
The paper is worthwhile and merits acceptance after addressing several matters below:
Thank you for your positive comment. It is really meaningful for us.
· The reactor wall temperature was fixed at 1290 K. Why such temperature and what is the meaning to externally heat a source of heat to maintain such wall temperature? The external temperature shall be determined by the heating power and heat loses to the environment.
Thank you for pointing out this mistake in the writing. The wall temperature is not known a priori and needs to be iterated for from the heat source due to reaction. As the solution is very sensitive to initial condition, an initial temperature at the wall is required to trigger the reaction numerically. Therefore, we have corrected Eq. 28 to Tinit = Twall. Moreover, we have added validation of temperature distribution with experimental data for low and high stoichiometry combustion.
· The introduction is adequate. However, it may benefit from a broader overview of catalyst-coated reactors to give a broader context. Examples of the works include DOI: 10.3390/catal8020058, 10.1016/j.cattod.2017.09.041, 10.1016/j.cej.2017.06.161, 10.3390/catal7120358
Thank you for the input given. Accordingly, the suggested literatures have been added to the revised manuscript.
· Section 2 seems rather too technical for a general reader. It may be worthwhile to summarise the section in a paragraph or two and put the original section into the appendix or supplementary.
Thank you for the suggestion. We agree that section 2 is little bit too technical for general reader. However, these details are required for the reader to follow the work presented in the manuscript. This is necessary if the reader would like to use the similar model for their work.
· Table 1 provides a dazzling array of various parameters which may benefit justification. Do the coils have the same length as the straight sections? Why are sizes (areas) of various channels different? Why platinum coverage is measured in kg*mol/m^2, not in kg/m^2 or mol/m^2
Yes, the length of the coil is similar to the straight counterparts. Similarly, the cross-section area of all geometries is the same. As for the unit of the platinum coverage, the correct unit is kmol/m^2. We apology for the mistake.
· Abbreviations and units in all the tables require clarification. Tables 2,4 in particular
The abbreviation and unit on Tables 2 and 4 (and other tables) have been checked and revised.
· Table 3 seem to present too minor details and does not warrant presence in the main text
We agree with the reviewer comment that value in Table 3 can be calculated from the data given in Table 1. Nevertheless, it was provided to give catalyst area for various configurations which is required to calculate the catalyst usage for reproducibility.
· The caption of table 4 does not explain what is the cross-section shape of the tube used
The caption has been revised accordingly.
· Figure 2. Simulations from the literature that do not fit the experiment only obstruct the view and does not contribute. It may be better to leave current simulations and experimental points only.
We do agree with reviewer comment. However, we feel it is better to include their numerical results as well to highlight the accuracy of the present model. We have also added validation of temperature distribution along the reactor channel in Figure 2b.
· Figure 4 and later. Please check units of concentration – these are not kmol, and neither mol. It must be something like mol/m^3
The unit has been checked and revised accordingly. It is kmol/m3
· Figure 5 and later. Why conversion is called “conversion rate”? The term “rate” implies the speed of change
We agree with reviewer comment. To avoid misunderstanding, the term has been change to “reactant conversion”
· Figure 6. Coated sections are written under the tube. It may be clearer to draw them on the outside of the figures to make it easier to understand
We have added the line in the walls to highlight the catalyst coating
· It may be good to do a minor proof-read and eliminate a few typos and eliminate or explain jargon such as “gassy mines”
Thank you. The manuscript has been proof read prior to resubmission the jargon has been explained in the revised manuscript.

Reviewer 2 Report
The manuscript deal with a very interesting topic.
Despite that the introduction should be improved in such points:
1) The authors just considered in the introduction the minimisation of methane harmful emissions by using catalytic combustion of CH4/air. They did not cited other possible technologies such as low NOx lean combustion, MILD combustion or oxycombustion.
Please cite such applications by referencing some literature studies :
- Jiang, X., Mira, D., & Cluff, D. L. (2018). The combustion mitigation of methane as a non-CO2 greenhouse gas. Progress in Energy and Combustion Science, 66, 176-199.
- Sorrentino, G., Sabia, P., de Joannon, M., Bozza, P., & Ragucci, R. (2018). Influence of preheating and thermal power on cyclonic burner characteristics under mild combustion. Fuel, 233, 207-214.
- Sabia, P., Sorrentino, G., Bozza, P., Ceriello, G., Ragucci, R. and de Joannon, M., 2019. Fuel and thermal load flexibility of a MILD burner. Proceedings of the Combustion Institute, 37(4), pp.4547-4554.
2) Please discuss better the formation of vortex flow in sections 3.2 and 3.3 . Is it a swirl or a cyclonic flow? Please report the swirl number of the system.
Several studies reported the utilisation of swirled or cyclonic flow in methane or hydrocarbons combustion. Please cite them:
- Schefer, R. W., Wicksall, D. M., & Agrawal, A. K. (2002). Combustion of hydrogen-enriched methane in a lean premixed swirl-stabilized burner. Proceedings of the combustion institute, 29(1), 843-851.
- Sorrentino, G., Ceriello, G., de Joannon, M., Sabia, P., Ragucci, R., Van Oijen, J., Cavaliere, A. and De Goey, L.P.H., 2018. Numerical Investigation of Moderate or Intense Low-Oxygen Dilution Combustion in a Cyclonic Burner Using a Flamelet-Generated Manifold Approach. Energy & Fuels, 32(10), pp.10242-10255.
3) Conclusions are too poor. Please enhance them.
Author Response
Response to Reviewer #2
We are very grateful for the positive reviews, comments, suggestions and the time spent reviewing our paper (Ref. No.: catalysts-481258). In the following, we have addressed all the issues raised by the reviewer in the revised paper. Below, we list the reviewer’s comments followed by our responses. The changes are highlighted in yellow in the revised manuscript:
The manuscript deal with a very interesting topic.
Thank you for your positive comment. It is really meaningful for us.
Despite that the introduction should be improved in such points:
· The authors just considered in the introduction the minimisation of methane harmful emissions by using catalytic combustion of CH4/air. They did not cited other possible technologies such as low NOx lean combustion, MILD combustion or oxycombustion.
Please cite such applications by referencing some literature studies:
- Jiang, X., Mira, D., & Cluff, D. L. (2018). The combustion mitigation of methane as a non-CO2 greenhouse gas. Progress in Energy and Combustion Science, 66, 176-199.
- Sorrentino, G., Sabia, P., de Joannon, M., Bozza, P., & Ragucci, R. (2018). Influence of preheating and thermal power on cyclonic burner characteristics under mild combustion. Fuel, 233, 207-214.
- Sabia, P., Sorrentino, G., Bozza, P., Ceriello, G., Ragucci, R. and de Joannon, M., 2019. Fuel and thermal load flexibility of a MILD burner. Proceedings of the Combustion Institute, 37(4), pp.4547-4554.
Thank you for the suggested literatures. The introduction has been revised by including the suggested literatures.
· Please discuss better the formation of vortex flow in sections 3.2 and 3.3 . Is it a swirl or a cyclonic flow? Please report the swirl number of the system.
Several studies reported the utilisation of swirled or cyclonic flow in methane or hydrocarbons combustion. Please cite them:
- Schefer, R. W., Wicksall, D. M., & Agrawal, A. K. (2002). Combustion of hydrogen-enriched methane in a lean premixed swirl-stabilized burner. Proceedings of the combustion institute, 29(1), 843-851.
- Sorrentino, G., Ceriello, G., de Joannon, M., Sabia, P., Ragucci, R., Van Oijen, J., Cavaliere, A. and De Goey, L.P.H., 2018. Numerical Investigation of Moderate or Intense Low-Oxygen Dilution Combustion in a Cyclonic Burner Using a Flamelet-Generated Manifold Approach. Energy & Fuels, 32(10), pp.10242-10255.
Thank you for the suggested literatures. The manuscript has been revised accordingly.
· Conclusions are too poor. Please enhance them.
Conclusion has been improved as suggested.

Reviewer 3 Report
This is an interesting work with the potential to appear in Catalysts. However, prior to publication, I have some comments that the authors should scrupulously implement into the revised manuscript.
1) Introduction - There is a lack in the literature survey on hydrocarbon (methane and propane) combustion in partially catalyst-coated monoliths. Thus, the authors should fill this gap. In particular, they should cite (and briefly discuss the core findings of) the following works where the opportunity of catalyst saving through selective coating of the monolith walls has been investigated via CFD simulations: Chemical Engineering Science, 116 (2014) 350-358; Catalysis Today, 242 (2015) 200-210 – in these works, only the external channels of the reactor, which are more exposed to heat loss to the external environment, were catalyst coated; Catalysis Today, 147 (2009) S156-S161 – in this work, the catalyst coating was present only in an initial part of the monolith channels (and not along their entire length).
2) Numerical methodology - The mesh independency of the numerical results should be supported by the comparison of the spatial profiles of the most important field variables obtained by varying the total number of cells.
3) Results and Discussion/Conclusions - The authors should better highlight the practical implications of their results.
I'm willing to review the revised manuscript.
Author Response
We are very grateful for the positive reviews, comments, suggestions and the time spent reviewing our paper (Ref. No.: catalysts-481258). In the following, we have addressed all the issues raised by the reviewer in the revised paper. Below, we list the reviewer’s comments followed by our responses. The changes are highlighted in yellow in the revised manuscript:
This is an interesting work with the potential to appear in Catalysts. However, prior to publication, I have some comments that the authors should scrupulously implement into the revised manuscript.
Thank you for your positive comment. It is really meaningful for us.
· Introduction - There is a lack in the literature survey on hydrocarbon (methane and propane) combustion in partially catalyst-coated monoliths. Thus, the authors should fill this gap. In particular, they should cite (and briefly discuss the core findings of) the following works where the opportunity of catalyst saving through selective coating of the monolith walls has been investigated via CFD simulations: Chemical Engineering Science, 116 (2014) 350-358; Catalysis Today, 242 (2015) 200-210 – in these works, only the external channels of the reactor, which are more exposed to heat loss to the external environment, were catalyst coated; Catalysis Today, 147 (2009) S156-S161 – in this work, the catalyst coating was present only in an initial part of the monolith channels (and not along their entire length).
Thank you for your kind suggestion. The suggested literature have been reviewed and cited accordingly.
· Numerical methodology - The mesh independency of the numerical results should be supported by the comparison of the spatial profiles of the most important field variables obtained by varying the total number of cells.
The suggested results have been added accordingly.
· Results and Discussion/Conclusions - The authors should better highlight the practical implications of their results.
The suggested discussion has been added to the revised manuscript.

Round 2
Reviewer 3 Report
The manuscript has been quite improved after revisions. However, I have still the following comments that that the authors should scrupulously implement into the revised manuscript prior to publication.
1) 1. Introduction, page 4, line 152 - Between the two sentences
“As the temperature of the inner channels increases due to heat conduction, the reaction in these uncoated inner channels is activated, thus allowing homogeneous reaction to occur throughout the entire monolith.”
and
“Selective coating applied in the catalytic combustion in a micro-combustor was examined by Di Benedetto et al. [31] using a two-dimensional CFD model.”
The authors should add the following sentence:
“On the basis of these numerical results [28-30], a novel partially coated monolith has been proposed and successfully tested by experiments [xx].”
with [xx] = Chemical Engineering Journal, Volume 259, 2015, Pages 381-390.
This will allow to better emphasize the key role played by CFD modeling and simulation in the management, design and operation of catalytic combustors.
2) 2.6 Numerical methodology - In Table 4, the authors give an example of results of mesh-independency tests. In particular, they give pressure drop and methane consumption (integral values) for the straight tube and the helical coil tube.
It would be nice to see, at least for one shape of the cross-section (for example, the square cross-section) the results of mesh-independency tests in terms of comparison of the spatial profiles of the most important field variables obtained by varying the total number of cells.
3) References - The authors should include this work in the final reference list: Chemical Engineering Journal, Volume 259, 2015, Pages 381-390.
I’m willing to review the revised manuscript.
Author Response
Response to Reviewer #3
We are very grateful for the positive reviews, comments, suggestions and the time spent reviewing our paper (Ref. No.: catalysts-481258). In the following, we have addressed all the issues raised by the reviewer in the revised paper. Below, we list the reviewer’s comments followed by our responses. The changes are highlighted in purple in the revised manuscript:
The manuscript has been quite improved after revisions. However, I have still the following comments that that the authors should scrupulously implement into the revised manuscript prior to publication.
Thank you for your positive comment. It is really meaningful for us.
· Introduction, page 4, line 152 - Between the two sentences
“As the temperature of the inner channels increases due to heat conduction, the reaction in these uncoated inner channels is activated, thus allowing homogeneous reaction to occur throughout the entire monolith.”
and
“Selective coating applied in the catalytic combustion in a micro-combustor was examined by Di Benedetto et al. [31] using a two-dimensional CFD model.”
The authors should add the following sentence:
“On the basis of these numerical results [28-30], a novel partially coated monolith has been proposed and successfully tested by experiments [xx].”
with [xx] = Chemical Engineering Journal, Volume 259, 2015, Pages 381-390.
This will allow to better emphasize the key role played by CFD modeling and simulation in the management, design and operation of catalytic combustors.
The suggested sentence and literature have been added to the revised manuscript.
· 2.6 Numerical methodology - In Table 4, the authors give an example of results of mesh-independency tests. In particular, they give pressure drop and methane consumption (integral values) for the straight tube and the helical coil tube.
It would be nice to see, at least for one shape of the cross-section (for example, the square cross-section) the results of mesh-independency tests in terms of comparison of the spatial profiles of the most important field variables obtained by varying the total number of cells..
The suggested results have been added accordingly as Figures 2 and 3.
· References - The authors should include this work in the final reference list: Chemical Engineering Journal, Volume 259, 2015, Pages 381-390.
The suggested literature has been added to the revised manuscript.
